# Colloid Carcinoma Arising in an Intestinal-Type Intraductal Papillary Mucinous Neoplasm with High-Grade Dysplasia Appearing as Signet-Ring Cells of the Pancreas by Serial Pancreatic Juice Aspiration Cytology: A Case Report

**DOI:** 10.3390/diagnostics13193123

**Published:** 2023-10-04

**Authors:** Mitsuhiro Tachibana, Takayoshi Hirota, Hideki Hamayasu, Yu Takeuchi, Kei Tsukamoto, Masahiro Matsushita

**Affiliations:** 1Department of Diagnostic Pathology, Shimada General Medical Center, Shimada 427-8502, Shizuoka, Japan; hamayasu@shimada-gmc.jp; 2Division of Pathology and Oral Pathology, Shimada General Medical Center, Shimada 427-8502, Shizuoka, Japan; 3Department of Gastroenterology, Shimada General Medical Center, Shimada 427-8502, Shizuoka, Japan; 4Department of Diagnostic Radiology, Shimada General Medical Center, Shimada 427-8502, Shizuoka, Japan

**Keywords:** claudin-18, cytopathology, colloid carcinoma (CC) of the pancreas, intraductal papillary mucinous neoplasm with high-grade dysplasia (IPMNHGD) of the pancreas, signet-ring cell, serial pancreatic juice aspiration cytological examination (SPACE)

## Abstract

We report a case of colloid carcinoma (CC) arising from an intestinal-type intraductal papillary mucinous neoplasm with high-grade dysplasia (IPMNHGD) of the pancreas, diagnosed with serial pancreatic juice aspiration cytological examination (SPACE). A rapidly growing intraductal papillary mucinous neoplasm (IPMN) in a 71-year-old Japanese man accelerated his hospitalization in our institute. Clinically, a large, ruptured pancreatic cyst was suspected. Cytologically, several mucin-positive signet-ring cells were scattered in the inflammatory, necrotic, or mucinous background. Signet-ring cells in cell block specimens were immunoreactive for MUC2, MUC5AC, maspin, S100P, and claudin-18. The final cytologic diagnosis was CC arising in an intestinal-type IPMNHGD with intraperitoneal penetration. The patient died two months after an explorative laparotomy. The cytologic diagnosis was achieved through SPACE, and the presence of signet-ring cells was characteristic. Anti-claudin-18.2-specific monoclonal antibody therapy will likely be used to treat patients with IPMNHGD in the future. This case highlights the diagnostic utility of SPACE, with particular emphasis on the characteristic presence of signet-ring cells. Furthermore, it anticipates the potential use of anti-claudin-18.2- specific monoclonal antibody therapy in the management of IPMNHGD patients.

A 62-year-old Japanese man was admitted to our hospital with respiratory distress, abdominal pain, and distension. The abdominal scans performed that it is to determine the current status of IPMN. Enhanced computed tomography (CT) and magnetic reso-nance cholangiopancreatography (MRCP) showed that the patient had a branch duct-type IPMN 22 mm in size. At the patient’s initial presentation to the hospital, the CT image showed a cystic lesion in the pancreatic tail (Figure 1a, arrow). The MRCP image showed a 22 mm cystic lesion in the pancreatic tail (Figure 1b, arrow). The mass increased in size over approximately three months, but the patient stopped visiting our hospital at his discretion. Nine years later, during hemodialysis at the Department of Nephrology, Shimada General Medical Center, for respiratory failure caused by acute renal failure, the now 71-year-old patient was found to have a pancreatic cystic mass on a CT scan and was referred to the Department of Gastroenterology for further examination and treatment. Laboratory examination showed serum levels of the tumor marker CEA of 6.4 ng/mL (standard value, ≤5.0) and CA19-9 of 56.6 U/mL (standard value, ≤37.0). MRCP showed that the main pancreatic duct was dilated to 8 mm in diameter, with a large ruptured pancreatic cyst of over 120 mm in diameter being suspected (Figure 1d), and an enhanced CT showed possible intraperitoneal penetration (Figure 1c). Although no high-risk stigmata were noted, the cyst diameter increased to ≥30 mm. The con-trast-enhanced cyst wall and main pancreatic duct ranging from 5 to 9 mm in diameter indicated worrisome features [1]. Therefore, we performed endoscopic nasopancreatic drainage (ENPD) and serial pancreatic juice aspiration cytological examinations (SPACE).

The final cytological diagnosis was CC arising from an intestinal-type IPMNHGD with intraperitoneal penetration. Thus, the patient was ruled ineligible for surgery and attempts were made to reduce the size of the pancreatic cyst with ENPD. However, the contents of the ENPD tube were viscous, and drainage was difficult. The patient’s abdominal pain and distension worsened, and respiratory distress developed. The surgeon suggested that the tumor cells derived from the pancreas penetrated the gastrointestinal tract. Open surgery was performed, during which a large amount of mucus accumulated in the abdominal cavity, and a small portion of the omentum was resected. The pathological diagnosis showed that the omental lesions were peritoneal dissemination of CC of pancreatic origin. Postoperatively, the patient was in poor condition. Respiratory distress developed following massive intra-abdominal mucus retention. Unfortunately, the patient died due to general deterioration two months postoperatively. Recently, there have been several case reports, mainly from Japan, in which the diagnosis of malignancy was confirmed by SPACE since Iiboshi et al. first reported its usefulness in preoperative diagnosis [4]. In pancreatic cancer, mucin-positive signet-ring cells, such as in the current case (Figure 2), appear to be associated with decreased cell adhesion [6]. The World Health Organization (WHO) reporting system for pancreaticobiliary cytopathology published in 2022 shows that IPMNHGD is characterized by small cells (similar in size to a 12 μm duodenal enterocyte) [3]. The present patient had signet-ring cells of 10 to 13 μm in size (Figure 2). In CC, tumor cells range from bland to signet-ring cells [3,5]. In our opinion, in pancreatic cytological diagnosis, it is possible to narrow the presumptive lesions to “CC,” “IPMN-derived invasive carcinoma,” “primary signet-ring cell carcinoma,” “anaplastic undifferentiated carcinoma (anaplastic carcinoma),” “neuroendocrine tumor with clear cell features,” and “the clear cell variant of the solid pseudopapillary neoplasm” by focusing on the appearance of the signet-ring cells or signet-ring cell-like cells [3,5,6,7,8,9]. Typically, in low-grade IPMNs, discohesive tissue fragments of irregularly arranged columnar cells with low nuclear/cytoplasm ratios are present [3]. When intestinal-type IPMNs become invasive, they often take the form of CC, as in the present case [2,3]. Only two cases of mucin-negative gastric cancer with signet-ring cell-like cells have been reported [10,11]. However, the presence of signet-ring cells was a distinctive cytological feature in the current case. To our knowledge, the present case is the first report of mucin-positive signet-ring cells in intestinal-type IPMNHGD evaluated by SPACE cytology. For cytological diagnosis of IPMNHGD, it is essential to observe stacked cell clusters and isolated scattering of atypical cells. Immunohistochemically or immunocytochemically, immunohistochemical staining for maspin and S100P is essential for diagnosing pancreatic adenocarcinoma [2,3,12]. Claudin-18 is often positive in ductal neoplasms, not only in invasive ductal carcinomas but also in IPMNs [13]. In the present case, immunohistochemically, the tumor cells were immunoreactive for maspin, S100P, and claudin-18 (Figure 3). Therefore, the patient was diagnosed with a CC of the pancreas. Sahin et al. revealed, in a study conducted in Japan using a monoclonal antibody (clone 43-14A), that claudin-18.2 showed a high rate of expression (87%) and was also expressed in IPMNs [14]. An ongoing clinical phase Ib/II trial (NCT04581473) or phase II study (NCT03816163) for metastatic pancreatic cancer has been reported that used chimeric antigen receptor T cells against claudin-18.2 in patients with advanced pancreatic adenocarcinoma [15,16]. Anti-claudin-18.2-specific monoclonal antibody therapy will likely be used in treating patients with pancreatic cancer, including those with IPMNHGD, in the future [16].

We report a case of signet-ring cells of CC arising from an intestinal-type IPMNHGD correctly diagnosed using SPACE cytology, which, to our knowledge, is the first to be published in the English literature. Signet-ring cells were one of the cytological features in this case. For the cytopathological diagnosis of CC, it is essential to observe stacked cell clusters and isolated scattering of atypical cells, and some floating atypical cells within the mucinous background show signet-ring features. Diagnosis based on Papanicolaou staining alone is difficult if only a few atypical cells are present. The specimen should be prepared and stained with maspin, S100P, claudin-18, or other stains to determine whether the cells are benign or malignant. In the future, anti-claudin-18.2-specific monoclonal antibody therapy will likely be used to treat patients with CC arising from an IPMNHGD.

## Figures and Tables

**Figure 1 diagnostics-13-03123-f001:**
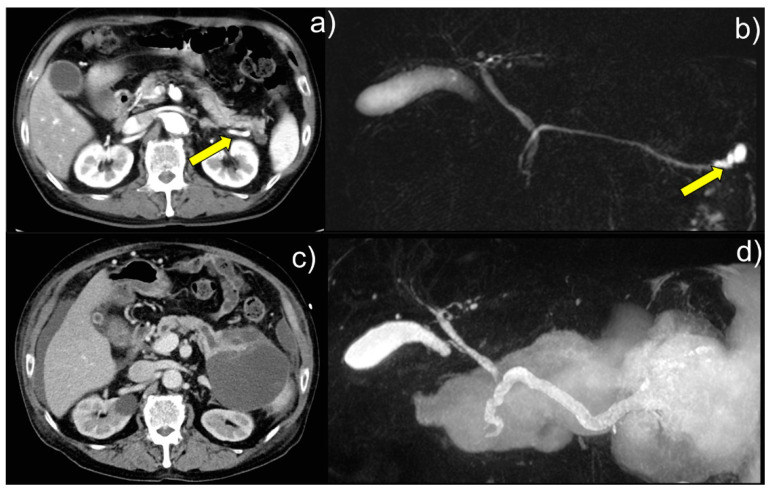
Enhanced computed tomography (CT) and magnetic resonance cholangiopancreatography (MRCP) showed that the patient had a branch duct-type IPMN 22 mm in size. (**a**) At the patient’s initial presentation to the hospital, the CT image showed a cystic lesion in the pancreatic tail (arrow). (**b**) MRCP image showed a 22-mm cystic lesion in the pancreatic tail (arrow). The mass increased in size over approximately three months but the patient stopped visiting our hospital at his discretion. (**c**) The CT image showed suspected intraperitoneal penetration, and (**d**) the MRCP image showed dilation of the patient’s main pancreatic duct to 8 mm in diameter. A large, ruptured pancreatic cyst (≥120 mm in diameter) was also suspected. Intraductal papillary mucinous neoplasms (IPMNs) represent approximately 1% of all pancreatic neoplasms and are cystic pancreatic lesions with mild to severe atypia. They are divided into main-duct IPMN, branch-duct IPMN, and mixed-type IPMN [1,2,3]. Understanding the characteristic cellular findings from cytological examination and referring to appropriate cytological diagnosis and histological subtypes is desirable [2,3]. Intestinal-type IPMNs comprise approximately 40% of IPMNs, and branch-duct IPMN has the highest frequency [4]. Colloid carcinoma (CC) of the pancreas is a rare subtype of pancreatic cancer, representing only 1–3% of the malignant tumors of the exocrine pancreas [5]. Pancreatic CC may arise in an intestinal-type IPMN or mucinous cystic neoplasm [5]. Cytologically, some malignant floating cells within the mucin of the CC show signet-ring cell features [5].

**Figure 2 diagnostics-13-03123-f002:**
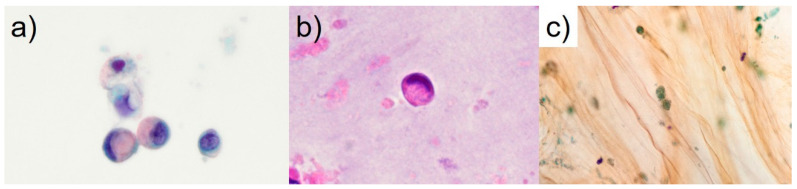
Cytological features of SPACE via ENPD tube. (**a**) Pancreatic juice cytology specimen shows many mucin-positive signet-ring cells. The nuclei of the tumor cells are hyperchromatic with prominent nucleoli. The signet-ring cells are mostly 10 to 13 μm in size (Papanicolaou stain, ×1000, oil, bar = 10 μm). (**b**) Many signet-ring cells are scattered in the cell-block specimen (hematoxylin and eosin stain, ×1000, oil). (**c**) Necrotizing cells are also seen in the inflammatory or mucinous background (Papanicolaou stain, ×600).

**Figure 3 diagnostics-13-03123-f003:**
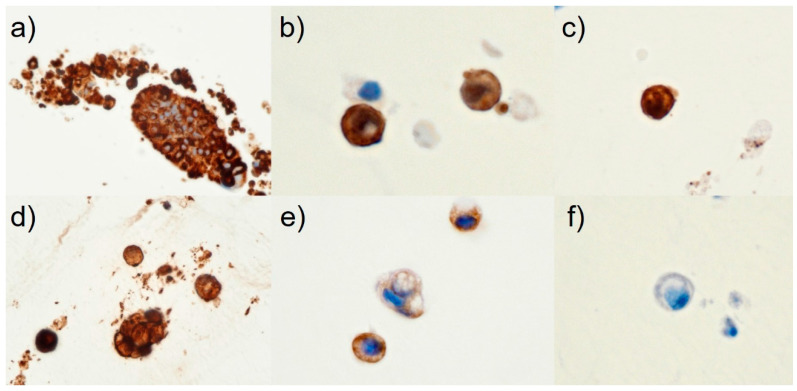
Immunohistochemical features in SPACE cell-block specimens obtained via ENPD tube (**a**–**f** ×1000, oil). Immunohistochemistry showed that the signet-ring cells were immunoreactive for maspin (rabbit polyclonal) (**a**), S100P (clone:16/f5) (**b**), claudin-18 (clone:66167-1-Ig) (**c**), MUC2 (clone: Ccp58) (**d**), and MUC5AC (clone: CLH2) (**e**), p53 (clone: DO-7) expression was negative (**f**).

## Data Availability

Not applicable.

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
