# Peer review of "Colloid Carcinoma Arising in an Intestinal-Type Intraductal Papillary Mucinous Neoplasm with High-Grade Dysplasia Appearing as Signet-Ring Cells of the Pancreas by Serial Pancreatic Juice Aspiration Cytology: A Case Report"

_diagnostics, 2023, doi:10.3390/diagnostics13193123_

Round 1
Reviewer 1 Report
Please take note of the attached file

--
Author Response
- In the abstract line 28 you might want to specify ‘postoperatively’, e.g. “after explorative laparotomy”, highlighting that curative pancreatic surgery was not an option.
- You might want to highlight your key point more aggressively, e.g. “This case highlights the diagnostic utility of SPACE, with particular emphasis on the characteristic presence of signet-ring cells. Furthermore, it anticipates the potential use of anti-claudin 18.2- specific monoclonal antibody therapy in the management of IPMNHGD patients.”
Thank you for your suggestion. I describe “The patient died two months after explorative laparotomy. The cytologic diagnosis was achieved through SPACE, and the presence of signet-ring cells was characteristic. Anti-claudin-18.2-specific monoclonal antibody therapy will likely be used to treat patients with IPMNHGD in the future. This case highlights the diagnostic utility of SPACE, with particular emphasis on the characteristic presence of signet-ring cells. Furthermore, it anticipates the potential use of anti-claudin 18.2- specific monoclonal antibody therapy in the management of IPMNHGD patients.”
The first paragraph (40-49) puts IPMN and colloid carcinoma in the context of pancreatic lesions. Because a strong emphasis of the work is cytology and the three types of IPMN are mentioned, you should correlate IPMN type and their most common histological types. The second paragraph (50-100) presents the case of a 72-year-old patient in chronological order and it provides a description of clinical and cytological imaging as well as the patient's history.
In lines 45-46, I describe “Intestinal-type IPMNs comprise approximately 40 % of IPMNs, and branch-duct IPMN has the highest frequency [4].”
- In line 55 you mention an increase in size over three months
- what follow-up imaging after the initial CT and MRI was obtained? Was the decision to stop follow-up made by the patient or was the lesion erroneously assessed and the follow-up terminated by the physician?
Thank you for your question. Nine years ago, after a branched duct-type IPMN was noted, the patient stopped visiting our hospital at his own discretion. Nine years later, the tumor had ruptured. The following was noted in Line 56 of the text ” the patient stopped visiting our hospital at his own discretion.“
- There is no mention of abdominal symptoms. Could you specify if the abdominal scan was obtained for diagnosis of respiratory failure or because of specific symptoms?
Thank you for your question. The abdominal symptom is abdominal pain and distension. The reason for performing a CT scan of the abdomen is to determine the current status of the IPMN. In line 51, I describe “abdominal pain and distension. The abdominal scans performed that it is to determine the current status of IPMN.”
- In line 63, please omit “of the patient”
Thank you, I omit that.
- In line 70 you cite the Fukuoka guidelines as “cause for concern”
- you might use the terminology of the guidelines and describe the contrast-enhanced cyst wall and the 9 mm duct as “worrisome features”
Thank you for your suggestion. In line 71, I replace “worrisome features” instead of “a cause for concern.”
- In line 93 I suggest “viscous” instead of “highly dense”.
Thank you for your suggestion. I replace “viscous” instead of “highly dense”.
- Did the patient die of surgical complications or due to general deterioration? The third paragraph (100-134) is concerned with general cytological features of pancreatic neoplasms and their classification.
Thank you for your comment. He died due to general deterioration. In line 101, I describe” Unfortunately, the patient died due to general deterioration two months postoperatively.”
- In line 114 replace “Generally speaking, however” with e.g. “Typically” The fourth paragraph summarises the paper and highlights the most noteworthy feature. Therefore I would stress their importance, e.g. “The presence of signet-ring cells was a distinctive cytological feature.”
Thank you for your comment. In line 114, I replace “Generally speaking, however” with “Typically”. And, in lines 120-123, “However, the presence of signet-ring cells was a distinctive cytological feature in the current case. To our knowledge, the present case is the first report of mucin-positive signet-ring cells in intestinal-type IPMNHGD evaluated by SPACE cytology.”
Reviewer 2 Report
Tachibana and colleagues present an interesting case report on a colloid carcinoma arising from an IPMN. They used SPACE to achieve clinical diagnosis. Although it is an interesting case, some further elaboration is needed. Why was the mass neglected? How did the patient do post-operatively? What was the reason of death?
Fine, some minor edits needed
Author Response
Why was the mass neglected?
Thank you for your question. Nine years ago, after a branched duct-type IPMN was noted, the patient stopped visiting our hospital at his discretion. Nine years later, the tumor had ruptured. The following was noted in Line 56 of the text ” The patient stopped visiting our hospital at his discretion.“
How did the patient do post-operatively?
Thank you for your question. Post-operatively, the patient was in poor condition. The following was noted in Line 99 of the text ” Post-operatively, the patient was in poor condition.“
What was the reason for death?
Thank you for your question. Respiratory distress developed following massive intra-abdominal mucus retention. the patient died two months postoperatively. The following was noted in Line 100 of the text ” Respiratory distress developed following massive intra-abdominal mucus retention. “